# Electrical Quantum Coupling of Subsurface-Nanolayer Quasipolarons

**DOI:** 10.3390/nano14181540

**Published:** 2024-09-23

**Authors:** Yihan Zeng, Ruichen Li, Shengyu Fang, Yuting Hu, Hongxin Yang, Junhao Chen, Xin Su, Kai Chen, Laijun Liu

**Affiliations:** 1School of Physics, Nanjing University of Science and Technology, Nanjing 210094, China; zengyihan@njust.edu.cn (Y.Z.); 921103860624@njust.edu.cn (R.L.); shengyu_fang@njust.edu.cn (S.F.); hxy123456@njust.edu.cn (H.Y.); chenjunhao0926@njust.edu.cn (J.C.); suxin@njust.edu.cn (X.S.); 2School of Mechanical Engineering, Nanjing University of Science and Technology, Nanjing 210094, China; huyuting@njust.edu.cn; 3MIIT Key Laboratory of Semiconductor Microstructure and Quantum Sensing, Nanjing University of Science and Technology, Nanjing 210094, China; 4College of Materials Science and Engineering, Guilin University of Technology, Guilin 541004, China

**Keywords:** multiferroic, bismuth ferrite, quasi-polaron, quantum dielectric physics

## Abstract

We perform dielectric and impedance spectrums on the compressively-strained ceramics of multiferroic bismuth ferrite. The subsurface-nanolayer quasipolarons manifest the step-like characteristic of pressure-dependent transient frequency and, furthermore, pressure-dependency fails in the transformation between complex permittivity and electrical impedance, which is well-known in classic dielectric physics, as well as the bulk dipole chain at the end of the dissipation peak.

## 1. Introduction

Bismuth ferrite (BiFeO_3,_ BFO) exhibits a diversity of physical phenomena, along with the potential for a number of device concepts, including beyond-CMOS logic gates, tunneling magneto-resistant spintronic valves, THz radiation emitters, ultrafast acoustic modulators, and linear electrooptical component. As a single-phase multiferroic compound, BFO exhibits coupled antiferromagnetic and ferroelectric properties at room temperature. The spontaneous polarization, *P_s_*, along the 〈111〉_pc_ pseudo-cubic direction interacts significantly with the internal strain. The formation of 180° domain walls to minimize depolarization energy results in reduced strain due to the release of non-180° (71° or 109°) domain walls, thereby stabilizing the crystalline solid at its lowest energy state. Therefore, the strain naturally exists in various forms, including bulk and low-dimensional structures, like ceramics, crystals, thin films, and nanodots. In ceramics, large compressive strain may drive more ferroelectric twinning to shape the irregular stripe patterns at nanometer scale within the elastic twin domains [1,2]. This may increase the quantity ratio of unusual 71° ± 1° grain boundaries and deformed 71° domain walls (~89° twin boundaries) [3]. It may curve the typical bridge-like 180° domain, and even pin the complexly-striped domain [4]. In crystals, it may, in a hydrostatic way, drive the multiple spin excitations of the noncollinear cycloidal state, to collapse into two excitations of a homogeneous antiferromagnetic state, which shows jump discontinuities at some of the ensuing crystal phase transitions [5]. Consequently, the *P_s_* is eliminated by the induced antipolar orthorhombic phase. While the compressive strain decreases the polarization, the tensile strain enhances the polarization in cross correlation with the increased activity of the electromagnon. This significantly suppresses the low energy magnon modes and, thus, induces the spin interstate transition [6]. In the low-dimensional form, it may make an evolving progression of domain, transitioning from strip patterns in films to flux-closure vortex or antivortex topology in nanodots [7,8]. Within the compressively-strained films, the significant strain gradient at the morphotropic boundaries between two structural domains can extrude the needle-shaped rhombohedral-like domain (R-domain) [9]. Simultaneously, it pins the tetragonal-like monoclinic domain, which is confined between two other tiled same-structure domains (T’-domain). The R-domain resembles an open circuit, enhancing the flexo-photovoltaic effect, whereas the T’-domain mimics a short circuit, leading to negative photo-conductivity. In short, the strain may alter crystalline, electron, and spin states, associated with excited phonons, magnons, and electromagnons, in a range from tens of nanometers to a few micrometers.

The domain represents a morphotropic phase characterized by finite-length dipole chains, particularly within the distorted perovskite structure (*R3c* space group) [10]. These dipole chains, in (001)-oriented crystals, terminate at a depth of 30–50 nm beneath the surface. Due to its significant period size of √3 × 6 × 6 μm, the dipole chain undergoes folding, curling, and winding to form various patterned nanodomains. It adopts the 〈111〉_pc_-oriented structure of “--O–Bi–O–Fe–O--”, where Bi and Fe ions are covalently bonded through O-*p* states. The large dipole of “O–Bi”, about 8.78 Debye units (D), alternates with the small dipole of Fe–O, about 3.16 D. Consequently, the dipole chains are correlated with the distorted structure. The lone pair of Bi-*6s* orbital in the main band position hybridizes with the lone pair of O-*2p* orbital above the <111> crystal plane, leading to a shift of Bi cation from its centrally symmetric position at about 0.61 Å. The hybridization also contributes to a long-range coupling of dipole chains for the ferroelectric polarization. The small displacement of Fe cation at about 0.22 Å distorts the FeO_6_ octahedron approximately in the same rotational direction as the polarization. The Dzyaloshinskii–Moriya and super-exchange interactions, induce a deviation in the magnetic moment of the Fe cation near the octahedron, resulting in a weaker G-type antiferromagnetic order. This strong correlation enables the multiferroic material to withstand thermal degradation up to a Curie temperature, *T_C_*, of 1100K, and a Néel temperature, *T_N_*, of about 640K.

In the nanolayer with a thickness of 30–50 nm positioned above the dipole chain bulk, a collective of quasipolarons at pseudo ground states are located for the depolarization field, with their pseudo-excited states, to capture various carriers for the charge compensation. Phenomenally, the dipole chains are for the bulk property, while the quasipolarons play a key role in the surface property [10], such as the three antiferromagnetic subdomains nested within one surface polarization domain of the quasipolarons in the subsurface quasi-polaron nanolayer of the (001)-oriented crystal, which is revealed by scanning-NV magnetometry [11]. The quasi-polaron carries a spin, because it is the quasiparticle of the unpaired 3*d^5^* electron in the Fe ion, which may show magnetic phenomena. In the ∼50-nm-thick (001) BFO films, different levels of strain modulated by (001) SrTiO_3_ and LaAlO_3_ substrate induce a rhombohedral-like and tetragonal-like crystal structure, respectively [12]. Only in the rhombohedral-like unit cell, the strain changes the tilting, rotation axis and magnitude of the FeO_6_ octahedra. By spin-lattice coupling, it makes the in-plane component of the antiferromagnetic order parallel with the applied magnetic field H. Under the H of 7 Tesla, the surface shows weak ferromagnetism, although the antiferromagnetic vector is confined to the (001) plane. As an alternative but simple physical modeling, the strain changes the lattice distortion of these quasipolarons, weakens the super-exchange interaction among the spins, and assists the field to more easily make the antiferromagnetic moment project onto the surface in order to be parallel to the H direction. The subsurface quasi-polaron nanolayer may be a structural base of the flexoelectric effect [13]. A mechanical strain gradient may increase or decrease the surface quasi-polaron polarization in antiferromagnetically dielectric [14] and ferroelectric [10] materials. In antiferromagnetic materials, Anderson initially models [15] a single d-electron in the presence of the diamagnetic lattice as the polaron-like quasiparticle (Figure 1) and, furthermore, in multiferroic BiFeO_3_ single crystals and ceramics, we name this electron the quasi-polaron [10]. Either in located states or in running states, it carries both the spin and the electric polarization of local lattice distortion. The many electron wave function of the single quasi-particle of momentum **k** and spin δ is [15]:(1)$$\Psi \left(k,\delta \right)={s_{k\delta }^{*}\Psi }_{0}$$

The *s*’s are a set of properly anticommuting fermion operators and details are shown in Ref. [15].

As one of the phases, polarons or quasipolarons often condense into small, twinned domains, and high-resolution transmission electron microscopy (HR-TEM) may reveal the local structure. For example, in Pr_0.5_ Ca_0.5_ MnO_3_, atomically resolved scanning transmission electron microscopy shows ordering, selected-area diffraction resolves the superstructure, together with column-resolved energy electron loss spectroscopy (which yields similar information to X-ray absorption spectroscopy); these measurements elucidate the ordering of Zener polarons [16]. However, the fingerprint of the quasipolarons is not traced by the TEM on ~380 nm and highly-strained BFO film [17] and the scanning TEM on one-unit cell-thick BFO films [18]. Scanning tunneling microscopy (STM) can spatially resolve the local structure at the atomic scale and reveals small polarons near oxygen vacancies in a strongly reduced rutile TiO_2_ (110) surface [19]. The STM often requires material with high conductivity and, therefore, scanning probe microcopy, including atomic force microscopy (AFM) and conductive AFM, has been performed on multiferroic insulator BFO films, which does not reveal the quasi-polaron [20]. Because of polarons’ and quasipolarons’ local and fine structures at subatomic scale, HR-TEM and other structural techniques [21] will be attempted in the next work as a challenge.

Hitherto, many electrical effect aspects of the subsurface-nanolayer quasi-polaron have remained unclear. Previous electrical characterizations are commonly reliant on equivalent circuit models within the frame of classic dielectric physics. Within the framework of quantum dielectric physics, the modeling of time-dependent density-functional theory has been applied to the optical properties of semiconducting and insulating systems, and ignores the lattice dynamics or any effects related to the coupling of electronic and lattice excitations, such as the polaron [22]. When the weak interaction or the van der Waals interaction is included, the modeling of local response approximation is used to calculate the distributed multipole polarizabilities of atoms in a molecule [23]. The modeling of the semiclassical Feibelman approach is obtained from many-body calculations performed in the long-wavelength approximation and neglects the nonlocality of the optical response in the direction parallel to the metal–dielectric interface [24]. However, these models often do not fully explain the Nyquist curve or provide a comprehensive microscopic view of such strong coupling at low frequencies. This motivates us to perform dielectric and impedance spectrums on compressively-strained ceramics to deepen the electric polarization understanding of subsurface quasipolarons, bulk dipole chains and intrinsic coupling. Our research addresses the fact that, because of the subsurface-nanolayer quasi-polaron, the electrical polarization is a macroscopic quantum effect [25] even in the ceramic bulk.

## 2. Materials and Methods

Ceramic disks have been reported on in our previous work [10]. After more than a decade of research, precise optimization of manufacturing parameters has enabled us to control the changes in the properties of multiferroic bismuth ferrite ceramics to an undetectable level. They are polished to the same thickness of 0.5 mm. Ag circular electrodes with diameters of 6. 0 mm are sputtered on one surface of the disks as top electrodes. The same material as that of the top electrode is sputtered to almost cover the other face as the bottom electrode. The home-made, modified and panoramic anvil (Figure 2a) was designed in accordance with the design chart (Figure 2b) and the material was mainly aluminum alloy. Pressure was applied by bolt torque, and the value is measured by hydraulic gauge. The disk is put between two parallel plate fixtures, which are connected to the room-temperature fixture in the impedance by copper wire leadings. Electrical spectrums use the Hewlett-Packard impedance/gain-phase analyzer (model 4294A, Agilent Co., Santa Clara, CA, USA) at room temperature. The a.c. voltage of sine waveform is applied with the amplitude ~ 0.5 V, in a frequency range from 10^2^ Hz to 10^7^ Hz.

When the disk is compressed in the sample cavity, two copper wire leadings from the parallel plate fixtures are connected to two probe heads of the room-temperature fixture in the impedance, respectively. The connection reliability is confirmed by measurement repeatability. The anvil is used to apply the compress up to 5.56 Mpa. The delicate pressure instruments and careful measurements make us apply and control strain in a precise manner to ensure uniform strain application and measure its effects accurately. Repeatability and consistency of measurement results are guaranteed by the high precision of the instruments, such as the frequency resolution of the analyzer with 1 MHz and the pressure resolution of the gauge with 0.02 MPa.

## 3. Results and Discussion

Figure 3a shows that the permittivity, ε′, is increased by the large pressure in all of the investigated frequencies, *f*. The strain compresses subsurface and bulk unit cells and, thus, increases the electrical polarization density of both subsurface-nanolayer quasipolarons and bulk dipole chains in the averaged volume. At the microscale, the strain is inhomogeneous due to the random grain orientations in the ceramics. The basin-shaped strain in the anisotropic subsurface lattices of certain grains forms local potential wells [26], trapping more quasipolarons to enhance the surface density of electrical polarization. Meanwhile, the downward-curved material–electrode interface drives the bending band of the electrode to penetrate across the surface and subsurface lattices, and deeper into the bulk lattice, accompanied with the punctiform potential well, to drag more branches of the dipole chains in order to be parallel to the applied a. c. electrical field. Notably, the density of the dipole chains at lower frequencies is larger than that of the quasipolarons at higher frequencies, leading to a greater increment in permittivity for bulk dipole chains below 10^6^ Hz than that of subsurface-nanolayer quasipolarons above 10^6^ Hz. In the vicinity of the coupling zone around ~10^6^ Hz (dashed line in Figure 3b) [10], a dissipation peak of loss tangent (tgδ) occurs at a frequency of approximately ~5.95 × 10^4^ Hz, which shows that a group of correlated dipole chains is at an energy level of about ~0.24 neV at room temperature to absorb and dissipate the energy of the applied electrical field. Then, they locate in a potential well far deeper than 25.4 meV, which is also the thermal energy at room temperature. With the increased strain, the peak height decreases, revealing that the dissipation becomes weaker. The unchanged peak position suggests that the compressive strain does not influence the Curie temperature (*T_c_*) of the ceramics, although it changes the *T_c_* in the Pb(Zr_1−x_Ti_x_)O_3_ (0.2  ≤  x  ≤  0.35) thin films [27], BaTiO_3_ nano-cubes [28] and SrTiO_3_ films [29]. Then, the decreased volume not only enhances the correlation strength of dipole–dipole within a dipole chain, but also intensifies those correlations among the dipole chains. The dipole chains are more united in concert with the increased strains, and behave like one body to make almost the same response to the electrical field. However, the value of the frequency dissipation peak is not linearly dependent on the strain and, thus, these correlations are not the linear function of spacing among the dipoles. The correlations may, to some extent, show the characteristic of Coulomb interaction, which is proportional to the square of the distance between two charges.

A clear observation of the binary dielectric species is manifested in the complex permittivity plot of ε′ and ε″, where ε″ is the imaginary part of complex permittivity, as shown in Figure 4a. The dot line of transient frequency, *f*_T_, separates the zone of the subsurface-nanolayer quasi-polaron from that of the bulk dipole chains. Without the pressure, *f*_T_ is about 10^6.27^ Hz, which reveals that the coupling energy between these two dielectric species is ~7.6 neV. With increased pressure, P, the *f*_T_ gradually moves into the larger value zone of ε′, which is consistent with the above analysis that the decreased volume simultaneously increases both the coupling strength and the electric polarization density. Figure 4b shows the pressure dependence of log(*f*_T_). With the increased pressure, it decreases from ~6.27 to ~6.23, then keeps constant in the pressure range from ~1 Mpa to ~3 Mpa, and finally shows the same step descent in the range from ~3.5 Mpa to ~5.5 Mpa. Such a decrease shows that the coupling energy is on separate energy levels, with 5.0 neV, 5.6 neV, 6.4 neV, 7.0 neV, and 7.6 neV. The discontinuous energy signs the quantum fingerprint. The states of subsurface-nanolayer quasipolarons and bulk dipole chains may be, to a large extent, quantum, for “birds of a feather flock together”. The decrease also shows that the shortened lattice increases the screening effect of the dipole chains, weakening the attraction aspect of the binary dielectric coupling. To our best knowledge, this decrease has not been reported before, which reveals that the electric polarization is a macroscopic quantum effect. Even in the superconducting phenomenon, which is also regarded as the macroscopic quantum effect, transport measurement under the similar condition does not show such a quantum characteristic [30]. When the in-plane electrical impedance is tentatively applied to the parent compound of pnictide superconductors, we do not find any quantum sign of interaction, which is believed to be the origin of high-temperature superconductivity [31].

The increased dipole density enhances the switching potential [32], strengthening the screening field and expanding the field range, thereby obstructing the path of the charge carriers. As shown in Figure 5a, the resistance, R, is increased below *f*_T_~10^3.84^ Hz, and the relative change, $∆R=R\left(P\right)-R(0)$, is larger than 10%. *f*_T_ reveals that, in the switching dynamic process, the charge carrier in the polarization current of the bulk dipole chains is different from the quasi-polaron in the leakage current of the subsurface-nanolayer quasipolarons. The bulk dipole chains contribute to the ferroelectricity, while the subsurface-nanolayer quasipolarons does to the reverse ferroelectricity [10]. And *f*_T_ shows that the coupling between two types of charge carriers at excited states is far smaller than that of those at ground states. However, its pressure-independence shows that the strain cannot break down the coupling, which is due to their dynamic electrical resistance. However, the ∆R shows negative values above *f*_T._ This suggests that the pseudo-excited-state waves of top and bottom subsurface-nanolayer quasipolarons overlap each other more and more with the increased pressure, providing a wider leakage pathway for charge carriers, or the pressure increases the density of the quasipolarons, but decreases the ranges of their screening field. Thus, the transporting pathway of charge carriers becomes “wider” and “clearer”. The different effects of the same pressure on these two screening fields may be due to the positive charge ion and the negative charge electron in the dipole chains, with only a negative charge electron in these quasipolarons, and their different lattice polarizations. The pathway seems solenoid-shaped. When the compress increases the number of solenoid tubes per radial length unit, the relative change of reactance, X, i.e., $∆X=X\left(P\right)-X(0)$, shows the larger positive value at all investigated temperatures, as shown in Figure 5b. Both ∆R and ∆X reveal the macroscopic aspect of electric polarization in the switching dynamic. They are not linearly dependent on the increased pressure, which also confirms that the related systems are coupled.

As a coupled many-body system in the subsurface nanolayer, the quasipolarons dissipate energy during the switching dynamic process of electric polarization in a way different from their behavior in the charge transporting process. In classical physics dielectrics, the single body is believed to put every effort into behaving as the charge carrier for the electric polarization switching. Therefore, there is a well-known transformation relation between the complex permittivity and the electrical impedance [33]. Based on this relation, the theoretical value of complex permittivity is transformed from the experimental value of the electrical impedance, which is relabeled as in the following:(2)$$\varepsilon \prime_{\mathrm{theoretical}}=-{\left[\omega C_{0}\left({R^{2}}_{\mathrm{experimental}}+{X^{2}}_{\mathrm{experimental}}\right)\right]}^{-1}X_{\mathrm{experimental}},$$
(3)$$\varepsilon {\prime \prime }_{\mathrm{theoretical}}={\left[\omega C_{0}\left({R^{2}}_{\mathrm{experimental}}+{X^{2}}_{\mathrm{experimental}}\right)\right]}^{-1}R_{\mathrm{experimental}},$$

In different pressure ranges of the quasipolarons in the subsurface nanolayer and the dipole chains in the bulk at certain frequencies, we introduce the relative change of complex permittivity to examine the transformation, as in the following:(4)$$\Delta \varepsilon^{\prime }=(\varepsilon \prime_{\mathrm{experimental}}-\varepsilon \prime_{\mathrm{theoretical}})/\varepsilon \prime_{\mathrm{theoretical}}\times 100\%,$$

(5)$$\Delta \varepsilon^{\prime \prime }=(\varepsilon {\prime \prime }_{\mathrm{experimental}}-\varepsilon {\prime \prime }_{\mathrm{theoretical}})/\varepsilon {\prime \prime }_{\mathrm{theoretical}}\times 100\%,$$
where ω=2πf is the angular frequency, and $C_{0}$ is the vacuum capacitance. The subscript “experimental” denotes the measured value of corresponding physical quantities. For the quasipolarons in the subsurface nanolayer, the value of Δε″ is, under the pressure of ~5 Mpa, as large as 400% at the frequency (Figure 6a), and the inherent coupling among these polarons fails the transformation relation. More polarons locate for the electrical polarization, but other, fewer polarons transport the charges. At the dissipation peak finish frequency of dipole chains in the bulk (Figure 6b), the different pressure incites the nature of dipoles at different pseudo-excited states and many-body couplings, leading to an unusual occurrence of energy dissipation. Therefore, the transformation also fails. At the frequencies of the dissipation peak and its onset (Figure 6c,d), the dipole chains are united in concert for the one-body response, and the transformation works. For the Δε′, it works for these two dielectric species, revealing a coupling strong enough for one-body electric polarization. Thus, the complex permittivity characterizes the localized states for electric polarization, while the electric impedance provides running states for transporting charges. Δε′ and Δε″ are used to differentiate between these behaviors. Although Anderson proposed an initial model based on quantum mechanics [15], the lack of experimental data on the quasipolarons in BiFeO_3_ prevents us from making a detailed model to elucidate the quasipolarons and their interactions with dipole chains. We hope to collaborate with theoretical physicists to model quasi-polarons, based on sufficient experimental results.

## 4. Conclusions

In summary, we discover the strain-induced change of dielectric and impedance spectrums in multiferroic bismuth ferrite ceramics. The quasipolarons in the subsurface nanolayer show that the transient frequency is step-like dependent on the pressure, and reveals that the coupling energy is quantum. The many-body characteristic of quasipolarons presents a difference in means of energy dissipation between the localized states for electric polarization and the running states for transporting charges. The quantum nature of coupled quasipolarons and dipole chains deepens our understanding of electrical phenomena in the multiferroic insulator. The subsurface quasi-polaron nanolayer with coupled electric polarization and anti-ferromagnetism may show potential for sensors, microwave devices, energy harvesting, photo-voltaic technologies, solid-state refrigeration, data storage recording technologies, and random access multi-state memories [34].

## Figures and Tables

**Figure 1 nanomaterials-14-01540-f001:**
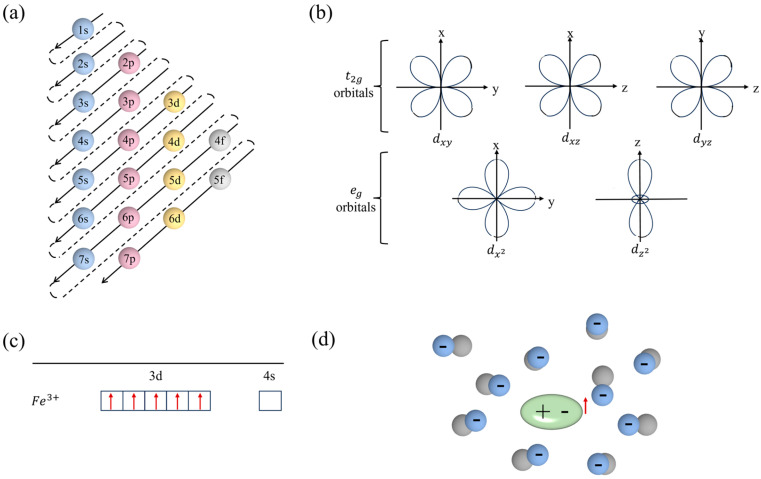
Electron configuration (**a**), crystal field splitting of d orbitals (**b**), spins of Fe^3+^−3d electrons (**c**) and the schematic illustration of the quasi-polaron in BiFeO_3_ (**d**).

**Figure 2 nanomaterials-14-01540-f002:**
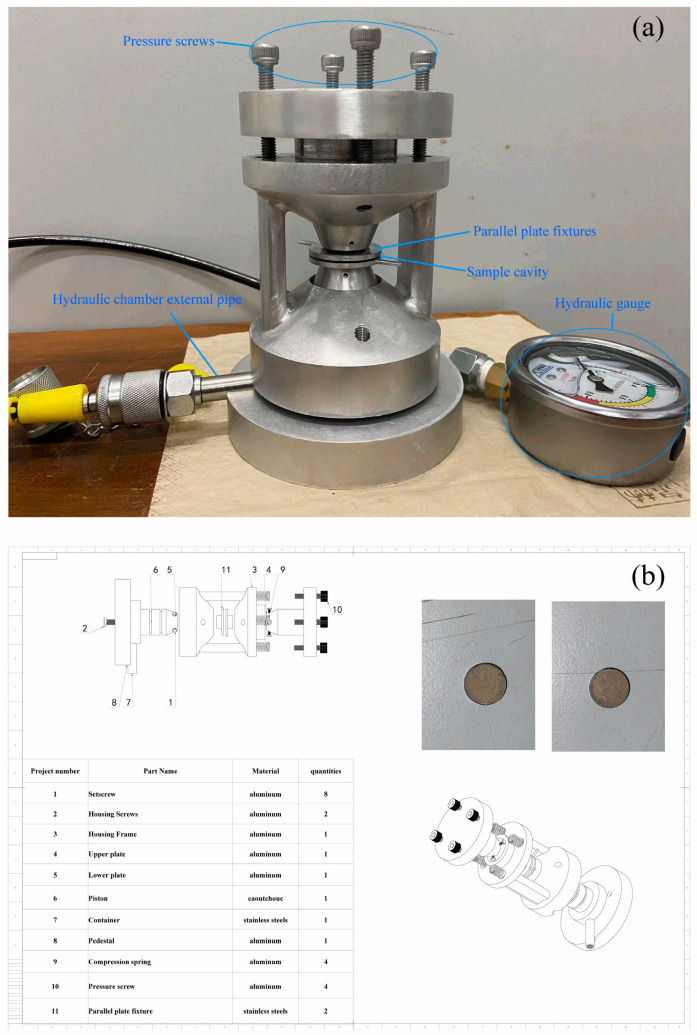
The homemade pressure anvil (**a**) and the design chart (**b**). The left (large) and right (small) insets show the unbroken disk with the silver bottom electrode and top electrode, respectively, after compressive strain is applied.

**Figure 3 nanomaterials-14-01540-f003:**
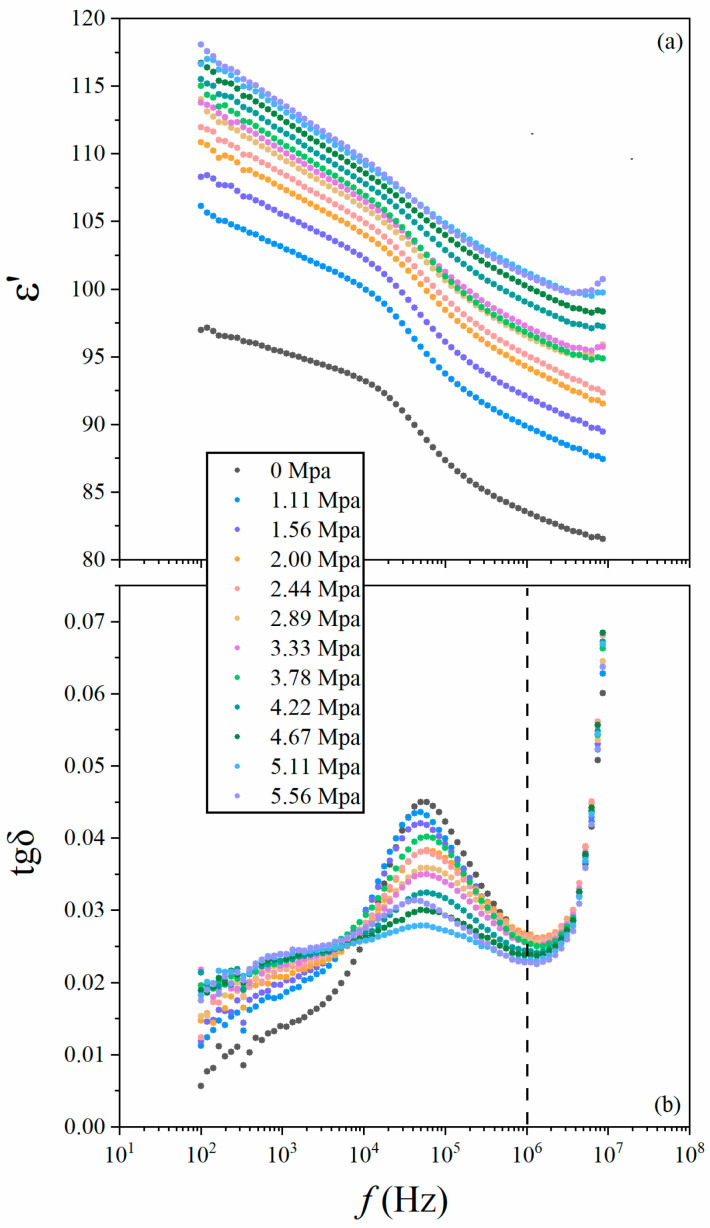
Frequency-dependent permittivity (**a**) and loss tangent (**b**) plots under pressure increased up to 5.56 Mpa.

**Figure 4 nanomaterials-14-01540-f004:**
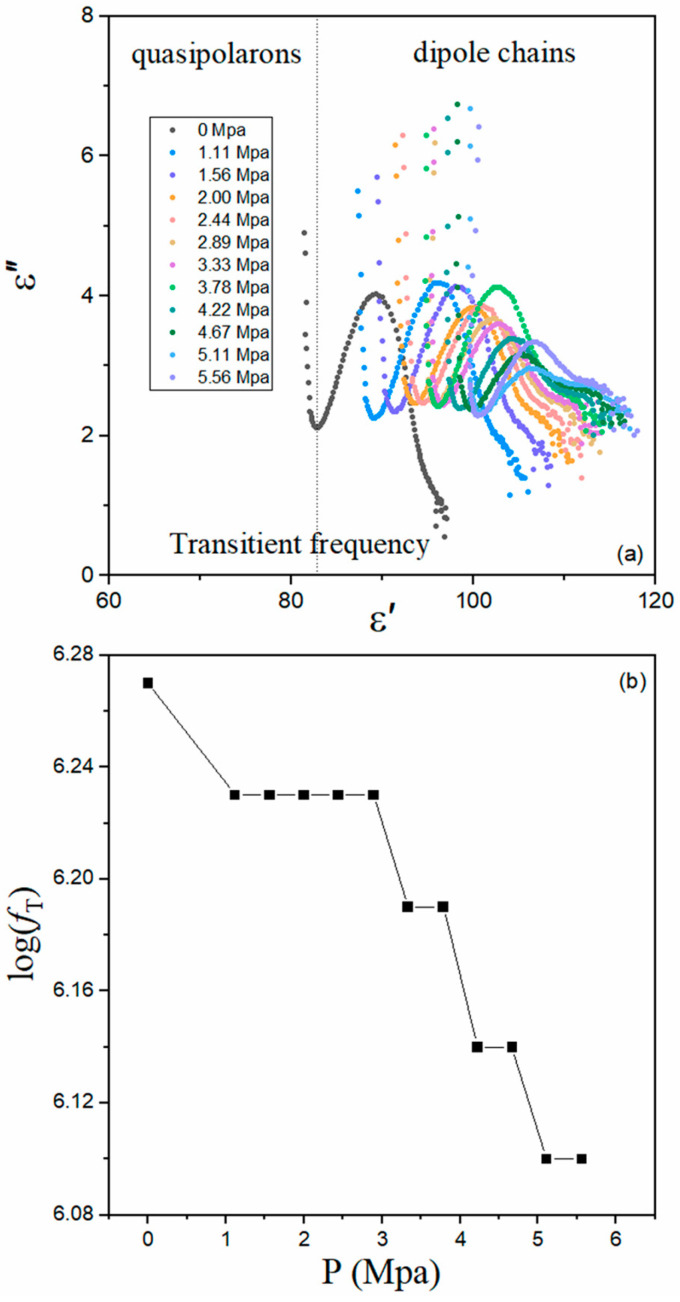
Complex permittivity plot under increased P (**a**) and pressure-dependent transient frequency (**b**). The dot line separates the response zone of the quasipolarons in the subsurface nanolayer and that of dipole chains in the bulk.

**Figure 5 nanomaterials-14-01540-f005:**
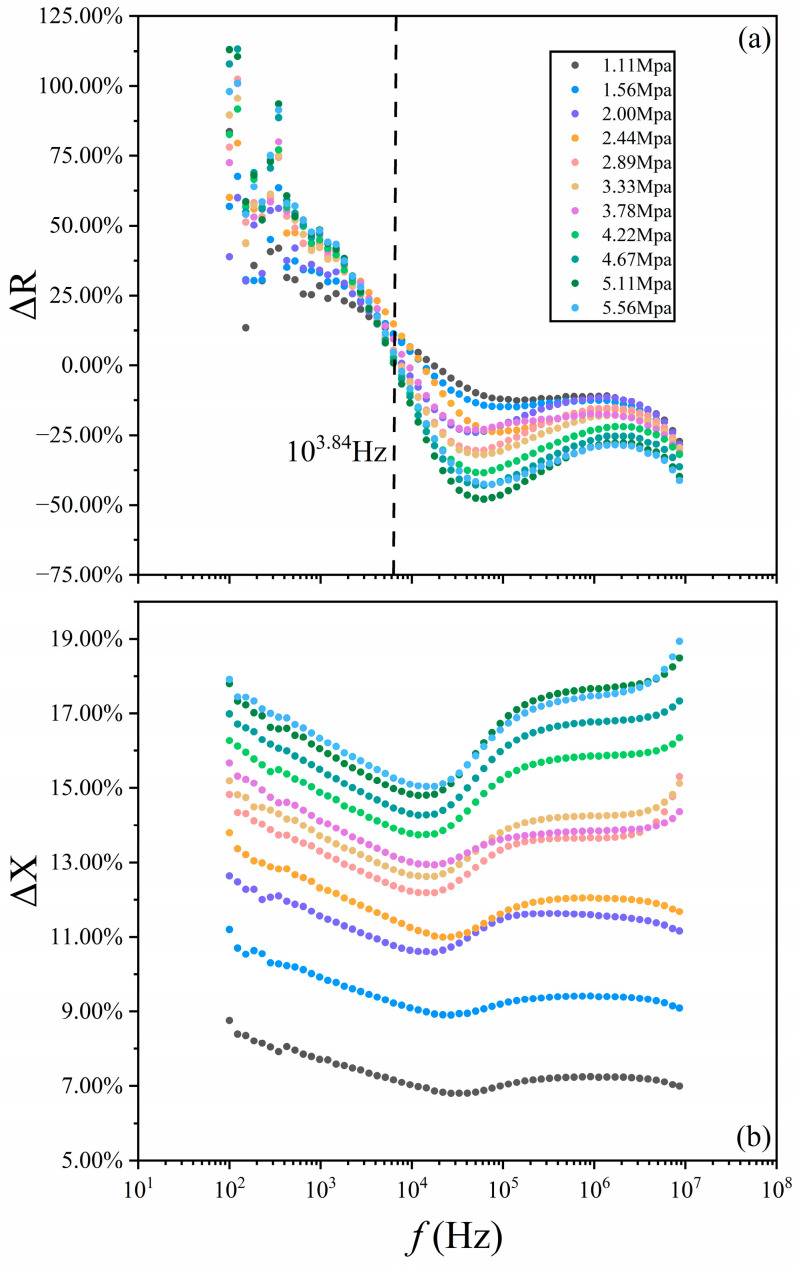
Relative changes in resistance (**a**) and reactance (**b**) plots under increased pressure.

**Figure 6 nanomaterials-14-01540-f006:**
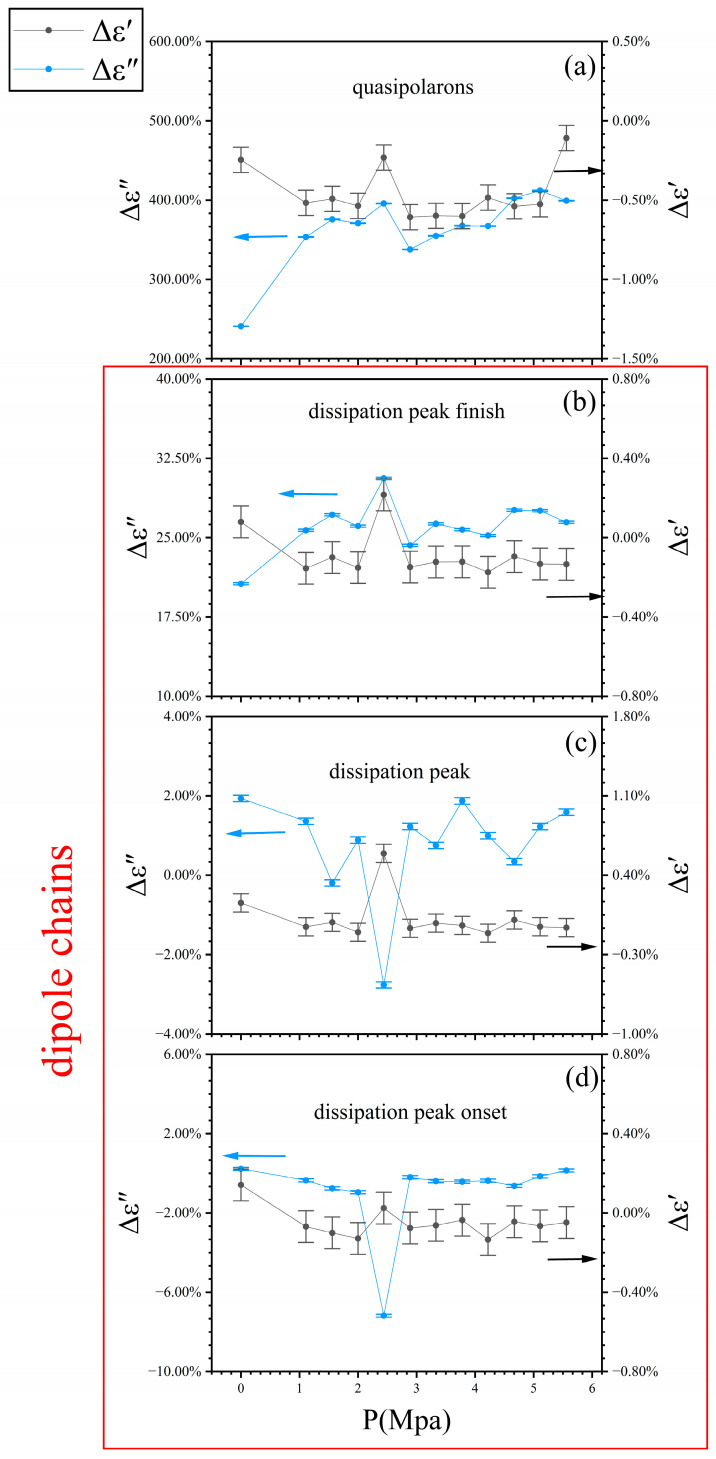
Relative change between complex permittivity and electrical impedance of the quasipolarons in the subsurface nanolayer (**a**) and the dipole chains in the bulk (**b**–**d**).

## Data Availability

The raw data supporting the conclusions of this article will be made available by the authors on request.

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
