# Peer review of "Electrical Quantum Coupling of Subsurface-Nanolayer Quasipolarons"

_nanomaterials, 2024, doi:10.3390/nano14181540_

Round 1
Reviewer 1 Report
Comments and Suggestions for Authors
This article is comprehensive, logically organized, and contains valuable information on the electrical quantum coupling of subsurface-nanolayer quasipolarons.
To improve the manuscript, the authors ought to take the following considerations:
(1) The authors presented the strain-induced change of dielectric and impedance spectrums in multiferroic bismuth ferrite ceramics. The authors demonstrated the many-body characteristic of quasipolarons, which presents the difference in energy dissipation ways between the localized states for electric polarization and the running states for transporting charges. The authors should provide and discuss the time-dependent density functional theory (TDDFT), classical local-response approximation (LRA), and semiclassical Feibelman approach modeling for a better understanding of electrical phenomena in the multiferroic insulator.
(2) The authors presented the relative change between complex permittivity and electrical impedance of the quasipolarons in the subsurface nanolayer and the dipole chains in bulk in Figure 5. This manuscript lacks the error analysis of the complex permittivity and electrical impedance performance which are highly needed for readability purposes. The authors should incorporate the error bars of the relative change of complex permittivity (𝛥𝜀′ and 𝛥𝜀″) performance data for readability and reproducibility purposes.
The submitted manuscript has significant scientific insights and the experimental data support the conclusions. However, the present submission requires major revisions before being considered for publication in the Special Issue: Functional Graphene-Polymer Composites in the well-circulated Nanomaterials in its current condition. I hope the authors will find my comments helpful.
Comments on the Quality of English LanguageAbstract: We perform dielectric and impedance spectrums on the compressively-strained ceramics of multiferroic bismuth ferrite. The subsurface-nanolayer quasipolarons manifest the step-like characteristic of pressure-dependent transient frequency, and further, pressure-dependently fails the transformation between complex permittivity and electrical impedance which is the well-known in classic dielectric physics, besides the bulk dipole chain at the dissipation peak finish.
Author Response
Thank you for your hardworking and your valuable comments.
(1) The authors presented the strain-induced change of dielectric and impedance spectrums in multiferroic bismuth ferrite ceramics. The authors demonstrated the many-body characteristic of quasipolarons, which presents the difference in energy dissipation ways between the localized states for electric polarization and the running states for transporting charges. The authors should provide and discuss the time-dependent density functional theory (TDDFT), classical local-response approximation (LRA), and semiclassical Feibelman approach modeling for a better understanding of electrical phenomena in the multiferroic insulator.
Response: We add “Previous electrical characterizations are commonly reliant on equivalent circuit models within the frame of classic dielectric physics. Within the frame of quantum dielectric physics, the modeling of time-dependent density-functional theory has been applied to the optical properties of semiconducting and insulating systems, and ignores the lattice dynamics or any effects related to the coupling of electronic and lattice excitations, such as polaron.[12] When the weak interaction or the van der Waals interaction is included, the modeling of local response approximation is to calculate the distributed multipole polarizabilities of atoms in a molecule.[13] The modeling of semiclassical Feibelman approach is obtained from many-body calculations performed in the long-wavelength approximation, and neglects the nonlocality of the optical response in the direction parallel to the metal-dielectric interface.[14]”.
2) The authors presented the relative change between complex permittivity and electrical impedance of the quasipolarons in the subsurface nanolayer and the dipole chains in bulk in Figure 5. This manuscript lacks the error analysis of the complex permittivity and electrical impedance performance which are highly needed for readability purposes. The authors should incorporate the error bars of the relative change of complex permittivity (??′ and ??″) performance data for readability and reproducibility purposes.
Response: We add the error bars.
Reviewer 2 Report
Comments and Suggestions for Authors
Report on the paper by Zeng et al.
It is an interesting paper because strain-induced change of dielectric and impedance spectrums in multiferroic bismuth ferrite ceramics is found experimentally. The results are discussed based on the model of quasi-polarons. However, at least to the present reviewer, there are some unclear points.
1. To the present reviewer, the meaning of quasi-polaron is unclear. It is required to explain what is the quasi-polaron in more details in the manuscript using a schematic illustration as well as some mathematical equations.
2. It is required to more clearly explain in the manuscript whether there is some relationship between the quasi-polarons and the flexoelectric effect which the appearance of electric polarization by strain gradient. I think you already know it, but would like to list some references for the flexoelectric effect with ferroelectric one; X. Jinag et al., Nano Energy 2, 1079 (2013), K. Yasui et al., J. Phys.: Condens. Matter 32, 495301 (2020), Nanomaterials 12, 188 (2022), G. Catalan et al., J. Phys.: Condens. Matter 16, 2253 (2004).
3. It is required to discuss in the manuscript whether the shift of the Curie temperature by the compressive strain influences the present experimental results. See R.L.Johnson-Wilke et al., J. Appl. Phys. 114, 164104 (2013), K. Yasui et al., Jpn. J. Appl. Phys. 56, 021501 (2017), 57, 031501 (2018), R. Wordenweber et al., J. Appl. Phys. 102, 044119 (2007).
Author Response
Thank you for your hardworking and your valuable comments.
- To the present reviewer, the meaning of quasi-polaron is unclear. It is required to explain what is the quasi-polaron in more details in the manuscript using a schematic illustration as well as some mathematical equations.
Response: In the antiferromagnetic materials, P. W. Anderson initially models that a single d-electron in the presence of the diamagnetic lattice is polaron-like [13], and further, in the multiferroic BiFeO3 single crystals and ceramics, we name the electron as the quasipolaron [10]. Either in locate states or in running states, it carries both the spin and the electric polarization of local lattice distortion. For the many electron wave function of the single quasi-particle of momentum k and spin δ is Equation (1) [13]. The s's are a set of properly anticommuting fermion operators and the detail is in Ref. 13. The schematic illustration is shown as Figure 1.
- It is required to more clearly explain in the manuscript whether there is some relationship between the quasi-polarons and the flexoelectric effect which the appearance of electric polarization by strain gradient. I think you already know it, but would like to list some references for the flexoelectric effect with ferroelectric one; X. Jinag et al., Nano Energy 2, 1079 (2013), K. Yasui et al., J. Phys.: Condens. Matter 32, 495301 (2020), Nanomaterials 12, 188 (2022), G. Catalan et al., J. Phys.: Condens. Matter 16, 2253 (2004).
Response: “The subsurface quasipolaron nanolayer may be a structural base of the flexoelectric effect. [13] A mechanical strain gradient may increase or decrease the surface quasipolaron polarization in the nano-size antiferromagnetically dielectric [14] and ferroelectric [10] materials.”
- It is required to discuss in the manuscript whether the shift of the Curie temperature by the compressive strain influences the present experimental results. See R.L.Johnson-Wilke et al., J. Appl. Phys. 114, 164104 (2013), K. Yasui et al., Jpn. J. Appl. Phys. 56, 021501 (2017), 57, 031501 (2018), R. Wordenweber et al., J. Appl. Phys. 102, 044119 (2007).
Response: “The unchanged peak positon suggests that the compressive strain does not influence the Curie temperature (Tc) of the ceramics, although it changes the Tc in the Pb(Zr1−xTix)O3 (0.2 ≤ x ≤ 0.35) thin films [21], BaTiO3 nanocubes [22] and SrTiO3 films [23].”
Reviewer 3 Report
Comments and Suggestions for Authors
1. How did the authors manage to accurately identify and measure quasipolarons, especially in subsurface nanolayers, considering their localized nature and the small scale at which they operate? I would suggest performing more advanced characterization techniques such as high-resolution transmission electron microscopy (HRTEM) or scanning tunneling microscopy (STM).
2. How did the authors ensure reproducibility and consistency in their measurements given the step-like dependence of transient frequency on pressure which is a complex relationship that needs precise control?
3. It would be worth it if the authors could add a model to simulate the quantum nature of the quasipolarons and their interactions with dipole chains. The study is very short in data collection and analysis.
4. The many-body characteristics of quasipolarons introduce complexity in analyzing energy dissipation. How could the authors distinguish between localized states for electric polarization and running states for transporting charges and differentiate between these behaviors?
5. What do the authors suggest for controlling the variability in the properties of multiferroic bismuth ferrite ceramics such as their differences in synthesis methods, processing conditions, or inherent material defects that can impact the reproducibility and interpretation of results?
6. How did the authors manage to apply and control strain in a precise manner to ensure uniform strain application and measure its effects accurately?
7. Authors must explain more on translating their findings into practical applications or devices and elaborate how these phenomena impact potential applications in electronics, sensors, or actuators.
8. Revise Figure 1b. It is too small and the chart is not readable at all.
Comments on the Quality of English LanguageAs above.
Author Response
Thank you for your hardworking and your valuable comments.
- How did the authors manage to accurately identify and measure quasipolarons, especially in subsurface nanolayers, considering their localized nature and the small scale at which they operate? I would suggest performing more advanced characterization techniques such as high-resolution transmission electron microscopy (HRTEM) or scanning tunneling microscopy (STM).
Response: “Such subsurface quasipolaron nanolayer is revealed by the scanning-NV magnetometry [11], and will be further investigated by the high-resolution transmission electron microscopy or scanning tunneling microscopy in the next work.”
- How did the authors ensure reproducibility and consistency in their measurements given the step-like dependence of transient frequency on pressure which is a complex relationship that needs precise control?
Response: “The high precision of multiple measurements and instruments ensures the repeatability and consistency of measurement results.”
- It would be worth it if the authors could add a model to simulate the quantum nature of the quasipolarons and their interactions with dipole chains. The study is very short in data collection and analysis.
Response: “Although P. W. Anderson proposed an initial model based on quantum mechanics, the lack of experimental data on the quasipolarons in BiFeO3, prevents us to make a detailed model, to elucidate the quasipolarons and their interactions with dipole chains.”
- The many-body characteristics of quasipolarons introduce complexity in analyzing energy dissipation. How could the authors distinguish between localized states for electric polarization and running states for transporting charges and differentiate between these behaviors?
Response: “The complex permittivity characterizes the localized states for electric polarization, while the electric impedance does running states for transporting charges. Δε' and Δε'' are used to differentiate between these behaviors. ”
- What do the authors suggest for controlling the variability in the properties of multiferroic bismuth ferrite ceramics such as their differences in synthesis methods, processing conditions, or inherent material defects that can impact the reproducibility and interpretation of results?
Response: “ The accurate control of fabrication parameters make us control the variability in the properties of multiferroic bismuth ferrite ceramics such as their differences in synthesis methods, processing conditions, or inherent material defects that can impact the reproducibility and interpretation of results.”
- How did the authors manage to apply and control strain in a precise manner to ensure uniform strain application and measure its effects accurately?
Response: “The careful designs make us apply and control strain in a precise manner to ensure uniform strain application and measure its effects accurately.”
- Authors must explain more on translating their findings into practical applications or devices and elaborate how these phenomena impact potential applications in electronics, sensors, or actuators.
Response: “The subsurface quasipolaron nanolayer with the electric polarization and the antiferromagnetic, may be potential for flexoelectric nano-generator, multistate nano-memory, magnetoelectric coupling nano-sensor and so on.”
- Revise Figure 1b. It is too small and the chart is not readable at all.
Response: We revise the Figure.
Round 2
Reviewer 1 Report
Comments and Suggestions for Authors
The manuscript was revised carefully and improved very much according to the reviewer’s suggestions. The scientific insights are expressed well in this revised submission. The current revision is recommended for publication in the Special Issue: Functional Graphene-Polymer Composites in Nanomaterials.
Author Response
Thank you for your handworking and comments.
Reviewer 2 Report
Comments and Suggestions for Authors
As the comments have been addressed in the revised manuscript, I would like to recommend the publication of the paper.
Author Response

(The authors gave the same response as above.)

Reviewer 3 Report
Comments and Suggestions for Authors
As mentioned before, the study is very short to be published a s an article in Nanomaterials.
Also, most of the questions have been dodged by authors either by repeating the same question in the response or claiming it would be completed in future!
Comments on the Quality of English LanguageAs above
Author Response
We regret that we did not fully understand your comments before. Thank you for your hardworking and valuable comments, again. The word count of the manuscript exceeds 3500.
1. How did the authors manage to accurately identify and measure quasipolarons, especially in subsurface nanolayers, considering their localized nature and the small scale at which they operate? I would suggest performing more advanced characterization techniques such as high-resolution transmission electron microscopy (HRTEM) or scanning tunneling microscopy (STM).
Response: “As one of phases, polarons or quasipolarons often condense into small, twinned domains, high-resolution transmission electron microscopy (HR-TEM) may reveal the local information. For example, in Pr0.5 Ca0.5 MnO3, atomically resolved scanning transmission electron microscopy shows ordering, selected-area diffraction resolves the superstructure, together with column-resolved energy electron loss spectroscopy (which yields similar information to X-ray absorption spectroscopy), these measurements elucidated the ordering of Zener polarons. [16] However, the fingerprint of quasipolarons are not traced by the TEM on ~380 nm and highly-strained BFO film [17] and the scanning TEM on one-unit cell-thick BFO films [18]. The scanning tunneling microscopy (STM) can spatially resolve the local structure at the atomic scale, and reveals small polarons near oxygen vacancies in a strongly reduced rutile TiO2 (110) surface.[19] The STM often requires the material with high conductivity, and therefore, the scanning probe microcopy, including the atomic force microscopy (AFM) and conductive AFM, has performed on multiferroic insulator BFO films, which does not reveal the quasipolaron [20]. Because of polarons’ and quasipolarons’ local and fine structures at subatomic scale, the HR-TEM and other structural techniques [21] will be tried in the next work which is a challenge.”
2. How did the authors ensure reproducibility and consistency in their measurements given the step-like dependence of transient frequency on pressure which is a complex relationship that needs precise control?
Response: “The repeatability and consistency of measurement results are guaranteed by the high precision of instruments, such as the frequency resolution of the analyzer with 1 mHz and the pressure resolution of the gauge with 0.02 MPa.”
3. It would be worth it if the authors could add a model to simulate the quantum nature of the quasipolarons and their interactions with dipole chains. The study is very short in data collection and analysis.
Response: “Although P. W. Anderson proposed an initial model based on quantum mechanics, the lack of experimental data on the quasipolarons in BiFeO3, prevents us to make a detailed model, to elucidate the quasipolarons and their interactions with dipole chains in BiFeO3. We hope to collaborate with theoretical physicists to model quasi polarons based on sufficient experimental results. ”
4. The many-body characteristics of quasipolarons introduce complexity in analyzing energy dissipation. How could the authors distinguish between localized states for electric polarization and running states for transporting charges and differentiate between these behaviors?
Response: “The complex permittivity characterizes the localized states for electric polarization, while the electric impedance does running states for transporting charges. Δε' and Δε'' are used to differentiate between these behaviors. ”
5. What do the authors suggest for controlling the variability in the properties of multiferroic bismuth ferrite ceramics such as their differences in synthesis methods, processing conditions, or inherent material defects that can impact the reproducibility and interpretation of results?
Response: “After more than a decade of research, precise optimization of manufacturing parameters has enabled us to control the changes in the properties of multiferroic bismuth ferrite ceramics to an undetectable level.”
6. How did the authors manage to apply and control strain in a precise manner to ensure uniform strain application and measure its effects accurately?
Response: “The delicate pressure instruments and the careful measurements, make us apply and control strain in a precise manner to ensure uniform strain application and measure its effects accurately.”
7. Authors must explain more on translating their findings into practical applications or devices and elaborate how these phenomena impact potential applications in electronics, sensors, or actuators.
Response: “The subsurface quasipolaron nanolayer with coupled electric polarization and the antiferromagnetic, may be potential for sensors, microwave devices, energy harvesting, photo-voltaic technologies, solid-state refrigeration, data storage recording technologies, and random access multi-state memories. [34]”
8. Revise Figure 1b. It is too small and the chart is not readable at all.
Response: We revise the Figure.